# Geospatial Distributions of Lead Levels Found in Human Hair and Preterm Birth in San Francisco Neighborhoods

**DOI:** 10.3390/ijerph19010086

**Published:** 2021-12-22

**Authors:** Chinomnso N. Okorie, Marilyn D. Thomas, Rebecca M. Méndez, Erendira C. Di Giuseppe, Nina S. Roberts, Leticia Márquez-Magaña

**Affiliations:** 1Department of Obstetrics, Gynecology, and Reproductive Sciences, University of California, San Francisco, CA 94115, USA; 2Department of Biology, San Francisco State University, San Francisco, CA 94132, USA; rmmendez@sfsu.edu (R.M.M.); edigiuseppe@berkeley.edu (E.C.D.G.); marquez@sfsu.edu (L.M.-M.); 3Departments of Epidemiology, and Biostatistics, Psychiatry and Behavioral Sciences, University of California, San Francisco, CA 94107, USA; Marilyn.Thomas@ucsf.edu; 4Department of Recreation, Parks and Tourism, San Francisco State University, San Francisco, CA 94132, USA; nroberts@sfsu.edu

**Keywords:** lead exposure, preterm birth, environmental racism, heavy metals, environmental pollution, urban contaminants, environmental justice, African American/Blacks, hair samples

## Abstract

In San Francisco (SF), many environmental factors drive the unequal burden of preterm birth outcomes for communities of color. Here, we examine the association between human exposure to lead (Pb) and preterm birth (PTB) in 19 racially diverse SF zip codes. Pb concentrations were measured in 109 hair samples donated by 72 salons and barbershops in 2018–2019. Multi-method data collection included randomly selecting hair salons stratified by zip code, administering demographic surveys, and measuring Pb in hair samples as a biomarker of environmental exposure to heavy metals. Concentrations of Pb were measured by atomic emission spectrometry. Aggregate neighborhood Pb levels were linked to PTB and demographic data using STATA 16 SE (StataCorp LLC, College Station, TX, USA). Pb varied by zip code (*p* < 0.001) and correlated with PTB (*p* < 0.01). Increases in unadjusted Pb concentration predicted an increase in PTB (β = 0.003; *p* < 0.001) and after adjusting for poverty (β = 0.002; *p* < 0.001). Confidence intervals contained the null after further adjustment for African American/Black population density (*p* = 0.16), suggesting that race is more indicative of high rates of PTB than poverty. In conclusion, Pb was found in every hair sample collected from SF neighborhoods. The highest concentrations were found in predominately African American/Black and high poverty neighborhoods, necessitating public health guidelines to eliminate this environmental injustice.

## 1. Introduction

Preterm birth (PTB) is well known as the most common cause of infant death and is the leading cause of disability-adjusted life years in the United States [1]. It is triggered by multiple pathological processes, including exposure to environmental factors [1]. Moreover, preliminary studies suggest that exposure to environmental toxins, such as lead (Pb), are associated with pregnancy complications, including PTB [2,3,4]. Notably, in San Francisco, the prevalence of PTB is significantly higher in low-income neighborhoods compared to high-income neighborhoods [5]. Social determinants of health conditions in the environments in which people are born, live, learn, work, play, worship, and age affect a wide range of health, functioning, and quality of life outcomes and risks. The range of personal, social, economic, and environmental factors influence the health status of individuals [2,6]. However, the interactions between the determinants of health such as race/ethnicity, poverty, environmental exposure to Pb and PTB rates are undefined. To address this gap in knowledge, this study examined environmental exposure of humans to Pb by measuring its concentration in hair and its correlations to PTB rates, race, and poverty for zip codes in San Francisco.

San Francisco (SF), like much of the world during the industrial era, relied on Pb to manufacture gasoline, paint, and pipes transporting water [3,7,8]. Lead-containing water pipes in older buildings, paint, and toxic waste from oil refineries are still present in the built-environment today in older cities like SF. For example, the SF Naval Radiological Defense Laboratory (NRDL) was operational from 1945 to 1974. NRDL activities contaminated soil, dust, sediments, surface water, and groundwater with petroleum fuels, pesticides, polychlorinated biphenyls, volatile organic compounds, radionuclides, and heavy metals such as Pb [9,10,11]. Superfund sites such as NRDL are found near low-income neighborhoods, including Bayview Hunters Point, which is predominantly African American/Black [9,10,11].

The burden of PTB on low-income and racially minoritized communities is apparent, and their location near environmental hazard sewage, waste, and hazard sites is clear [1,11,12]. Pervasive exposure to environmental toxins and ongoing racial inequities indisputably affect birth rates. PTB rates are disproportionately higher in racially minoritized populations and low-income neighborhoods compared to white and higher-income communities [5]. These trends are also found in SF neighborhoods where PTB rates exceed 8.0% of all live births and are higher in areas with greater poverty [5]. Low-income neighborhoods in SF not only suffer from significantly higher PTB rates, but the ratio of premature infants born to African American/Black, Latinx, and White mothers is 3:2:1, respectively [5].

In addition to the sites’ proximity to low-income communities, these populations have compounding exposure to health hazards by living in poor conditions. Poor living conditions is the reality for many non-White racial/ethnic groups which includes living in housing units built before the ban on Pb in the 1970s [7,9,11]. This includes neighborhoods such as Bayview Hunters Point, where the majority of African-American families reside in SF, where the soil contains unacceptable levels of toxic environmental chemicals, raising questions about the efficacy of the clean-up efforts used for their disposal [9]. It also raises questions about the level of human exposure to heavy metals in this neighborhood and others in SF.

Multiple studies have shown that hair readily uptakes some heavy metals [12,13,14,15]. Researchers postulate that the shuttling of Pb to hair may be an effective method that the human body uses to detoxify [13,15,16,17,18]. Thus, the concentration of Pb in hair is an indicator of long-term environmental exposure to this toxic heavy metal [12,15]. Prolonged exposure to Pb has been linked to a myriad of adverse health outcomes, including disruption of fertility and pregnancy [19,20]. Studies show that exposure to toxic chemicals in the environment can accumulate over time in the mother’s body and deposit in the fetus during pregnancy, ultimately resulting in PTB [1,19,20,21].

We evaluated the association between PTB and neighborhood-level Pb in the presence of other factors by using hair samples gathered from randomly selected neighborhood salons. According to survey results obtained from salon clients and staff, the hair gathered was from clients residing in these neighborhoods. We predicted that Pb concentrations in the collected hair would correlate with racial disparities in PTB outcomes for women residing in SF neighborhoods with high chemical waste sites and a greater prevalence of older homes.

## 2. Materials and Methods

This ecological-based observational study was designed to determine the relationship between PTB rates and Pb in hair collected from SF neighborhoods. Data collection consisted of a multi-method process including randomly selecting hair salons stratified by zip code, administering a brief in-house survey to record the clientele’s demographics, followed by isolating Pb from donated hair samples. Pb levels, demographic data, and publicly available public health data were analyzed using statistical software.

### 2.1. Participants and Study Design

The target population included 125 SF salons randomly selected across 26 zip codes (representing approximately 10% of hair salons in SF). The data generated from the successful collection of 109 donated samples from 72 salons (58%) across 19 zip codes (73%) of the target population (Figure 1). Zip codes were used as a proxy for neighborhoods because of the availability of PTB rate data [2]. Each participating establishment was categorized by service type (hair salons, hairstylists, and barbershops).

### 2.2. Biospecimen Collection and Processing Procedures

#### 2.2.1. Random Selection of Salons

The data scraping software, Octoparse (v. 7.0.2, Octopus Data Inc., Diamond Bar, CA, USA), was used to identify Beauty and Barber salons/shops in SF via two search engines, Google (general) and Yelp (local). To scrape for hair salon/beauty salon/barbershops, the two inclusion criteria were the following: (1) be located in one of the 26 SF zip codes with known PTB rates, and (2) be identified as a registered establishment by the SF Business portal or a hairstylist with a permit who rents a chair at a registered establishment. The 1012 establishments that met the inclusion criteria were used in Octoparse analyses by using salon data (name, neighborhood/area, address, zip code, service type).

To determine the average number of salons to sample per zip code, a statistical software, RStudio (v. 1.1, RStudio PBC, Boston, MA, USA), generated a random distribution of five salons in each zip code. This randomized quota sampling method helped account for changes in population density across the city and gave rise to aggregated values for each population by zip code. Ultimately, a random list of 125 salons in SF was generated and used as recruitment spaces for biospecimen collection (Figure 1). The intent was to collect samples from 10% (*n* = 125 salons) of the hair salon population in SF (*n* = 1012), with a sample size skewed between (Population size (*N)*: 1012, Confidence Interval (CI): 90%, margin of error (E): 10%) 64 salons and (*N*: 1012, CI: 99%, E: 10%) 143 salons. This study sampled at the lower end by collecting hair from 72 salons and 109 samples. Treasure Island, Alcatraz, and Farallon Islands were excluded since they did not meet the quota sampling of number of hair salons per zip code.

#### 2.2.2. Collection of Hair Samples

To confirm inclusion criterion, selected sample sites were contacted by phone to verify operating hours and if the business catered predominately to local residents and/or clients who work in the SF more than eight hours in the city (i.e., occupational exposure) [19,22,23].

Each client willing to participate was verbally consented and gave the authorization for the research team to collect hair samples. When no client was present, the business owner was consented. At minimum, the business owner was consented to record the demographics of clientele who, on average, visited the hair salon. This was necessary to record in order to account for the limitation that the hair collected belonged only to residents of that zip code.

After consent, laminated infographics designed for low literacy settings were disseminated to store owners, staff, and clientele. Each sheet provided information about environmental exposure and resources to increase awareness, facilitate conversations, and continue discussions with clientele around environmental exposure and possible health outcomes, consequently promoting trust between researchers and members of the community.

In addition, a 10-min interview in a survey format was administered to the consenting staff to understand the demographics of the general clientele population. The survey collected general information about the type of hair salon, cost of service, and locality of hair salon, sex, and race of customers used internally for verification. All responses were recorded on a digital device.

Then hair samples were collected by research staff filling a 250 mL container labeled high-density polyethylene with a mixture of trimmed hair from the floor or trash bin. The containers were stored at room temperature until processing. Survey data and biospecimens were de-identified to maintain privacy and confidentiality but were linked to the zip code where they were obtained for statistical analyses.

#### 2.2.3. Processing of Hair and Pb Measurement

De-identified hair samples labeled with a code linking them to the zip code were obtained, survey responses gathered during collection were processed, and Pb concentrations were measured. Acetone was used to wash hair samples donated from each salon to remove foreign impurities on the surface, before pulverizing in a Mixer Mill (MM 400, Retsch GmbH, Haan, Germany). Approximately 1.000 g ± 0.100 g of pulverized hair before digestion with a 1:1 ratio of 100% nitric acid and deionized water to isolate the inorganic metals that accumulate in the hair shaft after environmental exposure. This technique allowed for the measurement of Pb within the shaft of the hair and not the surface. The digested products were analyzed using an Agilent model 4200 Microwave Plasma–Atomic Emission Spectrometry (MP-AES, Agilent, Santa Clara, CA, USA) system optimized for measurement of Pb. Optimization for Pb quantification first required the calibration of the instrument response using Pb standards ranging from 0–10 parts-per-million (ppm). The Pb standards were prepared by gravimetric dilution of SPEX CertiPrep stock solutions (Spex SamplePrep LLC, Metuchen, NJ, USA) to generate calibration curves. The calibration curves were used to quantify Pb in the digested hair at the emission intensity of 405.781 nm.

At 405.781 nm, the limit of detection was 4 ppb (0.004 ug/g). Therefore, only measurements of Pb in hair greater than 4 ppb were deemed reliable and then used to determine Pb concentration. Pb concentration was quantified in all hair samples at levels above our limit of detection (Appendix A). These measurements were converted from emission intensity into the appropriate units (i.e., ug/g). Quality Assurance/Quality Control (QA/QC) procedures were implemented to ensure the reliability of the results, including analysis of appropriate Standard Reference Materials (SRMs/CRMs) to assess accuracy and analysis of triplicate samples to assess precision.

### 2.3. Data Analyses and Visualization

#### 2.3.1. Sources of Data and Statistical Analyses

Birth Data from the 2012 San Francisco Department of Public Health was used to measure PTB rate [2]. The 2016 American Community Survey was used to measure neighborhood percent poverty, percent with a bachelor’s degree, Gini index, percent African American/Black population, and foreign-born by zip code for assessment as confounders [24]. The association between Pb concentrations in inorganic extracts of hair and PTB in SF neighborhoods was determined using STATA 16 SE (StataCorp LLC, College Station, TX, USA). This software was used to conduct descriptive statistics, tests of mean differences (i.e., *t*-test, ANOVA), Pearson correlations, and linear regression models estimating the association between Pb and PTB adjusted for covariates significant at *p* < 0.05. The covariates adjusted for are the confounders identified from the 2016 American Community Survey mentioned above. Pb levels were also estimated using a 3-level ordinal variable (25th and 75th percentile cut-points) to confirm linearity. PTB rates and poverty percentages were used to measure statistical correlations.

#### 2.3.2. Data Visualization

ArcGIS geospatial mapping software (v. 2.2, Redlands, CA, USA) was used to visualize the distributions of Pb concentration and percent PTB, poverty, and African American/Black population by neighborhood (i.e., zip code). Each of these determinants formed a layer on a map of SF, allowing “hot spots” to be readily identified. Consequently, this data visualization approach is highly appropriate for place-based studies aiming to share environmental research results to improve public health.

## 3. Results

Overall, Pb levels varied by neighborhood and correlated with PTB rates. Increases in Pb concentration predicted an increase in PTB after adjusting for poverty. However, Pb concentration compared to PTB became non-significant after further adjustment for percent African American/Black population.

### 3.1. Pb Concentrations in Human Hair and Social Determinants Distribution in San Francisco

Pb amounts were detected in all samples, ranging from 1.5 ug/g to 40 ug/g, with a mean of 4.7 ug/g ± 2.5 using a 0.4 ug/g of Pb limit of detection reading from MP-AES instrument (Agilent, Santa Clara, CA, USA). For participating zip codes, the mean PTB percentage was 9.0% ± 2.4. The mean poverty rate was 13.3% ± 6.3, for bachelor’s degree 34.3% ± 7.7, for Gini index 50.2% ± 6.3, for African American/Black population 6.0% ± 6.2, and for foreign-born 32.3% ± 11.3 within the participating regions (Appendix A). Hairstylists reported that most of their clients were female-presenting and were from the same and/or adjacent neighborhoods as the establishment.

### 3.2. Poverty and PTB Distribution in SF

There was a significant positive correlation (R^2^ value = 0.23; *p* = 0.01) between neighborhood percent poverty and PTB rates (Figure 2, Table 1). Comparable to state-level and national-level averages (9.0% and 9.9%, respectively), the following neighborhoods had PTB rates above 9.0%: Treasure Island, South of Market, Bayview Hunters Point, Lake Merced/Merced Manor/Lake Shore, North Market/Hayes Valley/Tenderloin, St. Francis Wood/West Portal/Miraloma, Haight-Asbury/Hayes Valley, Visitacion Valley/Portola, Castro/Noe Valley/Corona Heights, Excelsior/Ocean View/Ingleside, Western Addition/Pacific Heights [5,22,25].

### 3.3. Relationship between Pb Concentrations and Preterm Birth Rate, Education, and Ethnicity

A pairwise analysis showed Pb concentration was correlated by neighborhood (adjusted R^2^ = 0.28; *p* < 0.001), and was weakly correlated with PTB rates (*r* = 0.27; *p* < 0.01), and with percent African American/Black population (*r* = 0.40; *p* < 0.001) (Table 2). These results were consistent with an unpaired *t*-test (two-sample *t*-test with unequal variances) that showed significant mean differences in Pb concentration by PTB, poverty, African American/Black population, foreign-born, bachelor’s degree, and Gini index (*p* < 0.001) (Appendix A). However, there was no significant correlation between Pb levels and percent foreign-born nor with poverty (*p* > 0.05).

Nested regression models showed that a one-unit increase in Pb concentration predicted a rise in PTB in the unadjusted model (β = 0.003; *p* < 0.001), and after adjusting for percent poverty (β = 0.002; *p* < 0.001). Results became non-significant after further adjustment for percent African American/Black population (β = 0.001; *p* < 0.16) (Appendix A). Similarly, a positive dose-response was found when Pb was assessed categorically in the unadjusted models (Appendix A): Compared to lower Pb levels (<3.0 ug/g), higher PTB was weakly associated with moderate levels of Pb (3–5.7 ug/g: β = 0.011; *p* = 0.067) and significant at the highest levels of Pb (>5.7 ug/g: β = 0.014; *p* < 0.05). Results also became non-significant after adjustment for percent poverty and African American/Black population (both β = 0.007; *p* ≈ 0.20), showing race is more indicative of high PTB rates than poverty level, further discussed.

### 3.4. Geospatial Representation of the Distribution of Pb in San Francisco

Figure 3 represents the geographical distribution of Pb, PTB rates, poverty rates, and African American/Black population density. Maps A–C show the distribution of PTB across zip codes in a blue base layer with darker tones reflecting higher PTB rates. Map A shows an overlapping dot distribution of poverty rates in purple. Map B shows an overlapping dot distribution of Pb concentration from hair samples in grey. Map C shows the overlapping dot distribution of Pb concentration and a heat map of African American/Black population density. Denser African American/Black population is reflected by the darker blue of the heat map.

PTB rates were concentrated South of SF (−122°25′ S) and most heavily in the Southwest (37°44′ W, −122°25′ S) and Southeast (37°44′ E, −122°25′ S). High poverty rates, high levels Pb exposure, and dense African American/Black population were concentrated East of SF (37°44′ E). These trends appear most heavily in Northeast (37°48′ E, −122°26′ N) and Southeast (37°44′ E, −122°26′ S) as shown by Figure 3, Map A, B, and C respectively.

## 4. Discussion

### 4.1. Primary Findings

The primary finding is that the majority of hair samples obtained from different SF neighborhoods contained a concentration of Pb within the hair shaft accumulated over time. The highest Pb exposure was measured in Eastern SF, predominantly in the Southeast neighborhoods, where a large population of African American/Black resides, and PTB rates soar. Additionally, PTB rates were higher in neighborhood pockets encompassing hazardous waste sites, high poverty rates, and racial/ethnic communities. This important study highlights the need for more translational community participatory research to help bridge the gap between basic science and epidemiological research to illustrate a clear picture of health disparities.

This is the first ecological study to measure Pb exposure in San Francisco (SF) using a novel, low-invasive hair sample approach to investigate correlations between Pb concentration, PTB, and neighborhood determinants of health. The use of hair salons as a focal point for community engagement allowed us to be innovative in exploring the knowledge gap in understanding correlations between human exposure to heavy metals in SF neighborhoods and PTB across multiple levels. The average Pb concentration detected in hair samples (*n =* 108) was 4.7 ug/g (±2.5) from 72 hair salons spread across 58% of SF county. When Pb concentrations were aggregated by zip code, Pb concentrations were significantly and positively associated with PTB rates and African American/Black population density. Furthermore, higher poverty percentages modestly trended with higher PTB. Additionally, negative correlations were found between Pb concentration and both bachelor’s degree attainment and Gini index, but they were statistically non-significant and thus weakly correlated (Table 2).

Overall, results of this present study show that high Pb exposure was found in San Francisco neighborhoods with larger numbers of African American/Blacks and higher preterm birth rates. Predominantly African American/Black communities experience historical and ongoing neglect and divestment that contribute to disproportionate exposure to high-risk factors, including poverty and environmental toxins leading to health disparities. For example, African American/Black families are more likely to live in poor areas with dilapidated housing and/or old infrastructure (e.g., paint/pipes) increasing their risk for environmental toxin exposure [23]. This further explains why we found a moderate correlation between poverty and percent African American/Black population (*r*
*=* 0.52). Together our findings suggest that interventions aimed to address the negative impact that Pb exposure may have on PTB should target poor African American communities.

The association between PTB and Pb became non-significant in the linear regression models adjusted for poverty when percent African American population was added. It seems that the effects of poverty reduced in the presence of race in dense concentrated areas with high PTB. As seen geospatially, race becomes a determining factor for PTB. This finding aligns with the effects of racism on African American/Black communities, which prevent them from attaining their highest level of health. For example, institutional racism have placed policies and practices that have confined African American/Black populations to poor areas and neighborhoods containing toxic waste. Thus, toxic metals such as Pb, can potentially add to stressors at both the neighborhood- and individual-level, interacting with other social determinants of health increasing the risk for PTB and the likelihood of mother/infant mortality and morbidity in conjugation of race or poverty level.

### 4.2. Perspective of Previous Studies

Prior research shows that detrimental health outcomes related to Pb exposure date as far back as 1911, following the industrial age of the United States [26]. At that time, Pb was a component added to gasoline, paint, and commercial pipes before an industrial ban was enacted in the 1970’s [3,7,8]. This ban effectively reduced new exposure but did not guarantee the eradication of existing Pb circulating in the lived environment. Moreover, neighborhoods with hazardous waste sites were environmentally burdened by additional sources of lead. Thus, environmental exposure to Pb in these neighborhoods may occur through proximity to leaching of Pb from contaminated hazardous waste sites and neglected housing [27,28].

In SF, more than 80% of the homes built before 1979 are coated with Pb-based paints [23]. Pb-containing paint becomes hazardous when it starts to peel, crack, or is otherwise disturbed by repairs or renovation projects and is inhaled, ingested, and absorbed over time. In fact, a study conducted from 2008 to 2012 in SF using blood biospecimens from children showed increased Pb levels from 6.6% to 11.8% as city renovation projects increased. Many of these children lived in poor neighborhoods [23]. Low-income, densely populated, and racially minoritized SF neighborhoods with higher Pb levels are also proximal to EPA exposure sites (Figure 3).

The connection among high-risk communities, waste sites, and Pb contaminated housing can be explained by historical neighborhood redlining practices also linked to adverse birth outcomes [29,30]. Redlining—the state-sanctioned ranking of neighborhood desirability for lending purposes—created clustering of racial groups in cheaply constructed, poorly maintained, and underfunded housing and community buildings. These previously ostracized and infrastructural neglected neighborhoods are currently undergoing renovation projects exposing the community to debris and potentially heavy metals [23,29,30].

Like other heavy metals, Pb exposure is associated with chronic stress, oxidative DNA damage, and elevated prenatal stress during pregnancy [31]. During pregnancy, toxic chemicals that accumulate in the mother’s body over time can deposit in the fetus leading to low birth weight and PTB [1,18,20,21,32]. PTB and other adverse health outcomes affecting the mother and fetus are a consequence of chronic inflammatory responses caused directly and indirectly by environmental chronic stressors such as Pb [31,33,34,35,36]. Pb is also associated with disruption of the reproductive system in both male and female sexes [20,21]. Examples in males include abnormal spermatogenesis, chromosomal damage, infertility, abnormal prostatic function, and changes in testosterone [27,37]. Examples in females include susceptibility to infertility, miscarriages, premature membrane ruptures, preeclampsia, pregnancy hypertension, and premature delivery [27,37].

### 4.3. Implications of Findings

Pb exposure is detrimental to women’s reproductive health and disproportionately affects marginalized individuals, namely, those who live below the poverty line and in low-income areas near poorly maintained industrialized facilities. Waste sites, construction sites, and older buildings are all potential sources of exposure to continuous Pb leaching. In fact, their geographical overlap in SF indicates how disenfranchised residents are exposed to heavy metals and why they are at higher risk for reproductive health disparities [5,25]. For example, the geospatial data (Figure 3) shows that Eastern SF has EPA sites located approximately five miles away from areas with higher levels of factors significantly associated with PTB in this study: Pb concentrations in hair, high poverty/low-income, and a dense African American/Black population.

Poverty, high Pb exposure, and a large population of African American/Black seem to be factors that permeate the Bayview Hunters Point neighborhood. Many of its residents live in public or subsidized housing that is often not well maintained [10]. It is home to low-income and high-minoritized populations and is densely populated by industrial companies, toxic waste sites, and other sources of contamination. For example, this neighborhood was home to a federal Superfund site in the Hunters-Point Naval Shipyard (NRDL), which continues to house the PG&E Hunters-Point Power Plant, and the largest sewage treatment plant in SF. NRDL activities contaminated the soil, surface and groundwater with heavy metals, among other toxins [9,10,11]. As a result, both residents and the environment continue to be impacted by both stationary and mobile pollution sources, including radioactive and toxic contamination at the superfund site and nearby waterfront [10]. The most recent CalEnviroScreen report concluded that there is continued leaching of these toxins and that the impact on residential health has contributed to a higher prevalence of asthma, cardiovascular disease, and low birth weight over the years [23].

The health implication of long-term exposure to environmental toxins includes poor birth outcomes, mental retardation, infertility, and disability-adjusted life-years in the US. These adverse health outcomes affect SF’s workers and residents, making this a considerable and serious public health and environmental justice concern [38].

### 4.4. Future Directions

Further studies should regularly monitor human exposure to Pb and other common heavy metals by measuring its levels in workers and residents using non-invasive methods like the collection of hair. The results obtained by determining the concentration of Pb in the collected hair should be shared with stakeholder communities using online geospatial tools. These tools allow for readily understandable visualization of the data. Additionally, healthcare policies should be changed and resources made available to facilitate secondary prevention methods for Pb exposure. Specifically, toxic screening panels should be added to annual routine check-ups, especially in children, high-risk women of childbearing age, and those living in areas near heavy construction and or waste disposal sites. Additionally, resources should be made available to educate the community on signs of heavy metal toxicity and recommended steps to address the effects exposure.

Heavy metal exposure is not unique to SF. Flint is a city in Michigan where Pb-tainted water sources created a cascade of health problems affecting community health [39,40]. Similar health disparities have appeared in older metropolitan cities in states like Pennsylvania, New York, and Iowa with elevated levels of environmental toxins associated with adverse symptoms in children [5,20,23]. Consequently, at the national level, future directions include routine evaluation of current policy and regulatory practices to identify successful policy and interventions, and strengthen enforcement against those who neglect proper hazardous waste disposal practices.

#### Limitations and Delimitations

This current ecological study design and non-purposive sampling preclude causal interpretation and generalizability of results beyond the 26 zip codes studied. Further, the small sample size increased the likelihood of committing a Type II error. A key delimitation of this study is that out of all the salons and barbershops in SF, 125 were randomly invited to participate to capture a sample of the population. Sampling of hair from local businesses increased randomization, but it did not guarantee that all the hair samples collected in one business only belonged to residents of that zip code. The collection of hair samples used was considered less invasive and a more accessible method of collecting biospecimens from the public, especially those who are hesitant to engage in research. In future studies, these findings could be corroborated by measuring heavy metal levels in both blood and hair to have both real-time and cumulative data. A limitation of the current approach regarding PTB is the inability to link Pb concentrations obtained at the neighborhood level to pregnancy. This limitation could be overcome in future studies by recruiting pregnant women from SF neighborhoods to donate biospecimens to measure exposure to environmental chemicals during pregnancy, after birth, and in their child.

## 5. Conclusions

In this study, environmental exposure of Pb in SF zip codes and its associations with determinants of health was examined by race, poverty and PTB rates. Leveraging the social-ecological model (Social-ecological models recognize individuals as embedded within larger social systems and describe the interactive characteristics of individuals and environments that underlie health outcomes [41]), we examined environmental exposure of Pb in neighborhoods by aggregating hair samples from multiple individuals within each zip code. Using zip codes as a proxy for municipality allowed us to analyze the interactions among Pb exposure, PTB rates, and determinants of health at the community level.

Pb levels were found in human hair at the individual-level that can plausibly be explained through environmental exposure. The Pb measured was geospatially correlated with higher rates of preterm birth, poverty, low-income, and higher density of African American/Blacks in San Francisco. Notably, San Francisco local businesses in hard-to-reach communities were engaged and were the primary sites of biospecimen donation used in this analysis. The findings are unique because of the translational community-engaged study design, successful recruitment of racially underrepresented study participants, and the visualization of significant interactions with geospatial mapping.

This important study highlights the need for more translational community-engaged research to bridge the gap between traditional science and epidemiological research to illustrate a clear picture of health disparities. In particular, this study shows how social determinants including toxic environmental exposures exacerbate health disparities at the individual and community level [9,35].

## Figures and Tables

**Figure 1 ijerph-19-00086-f001:**
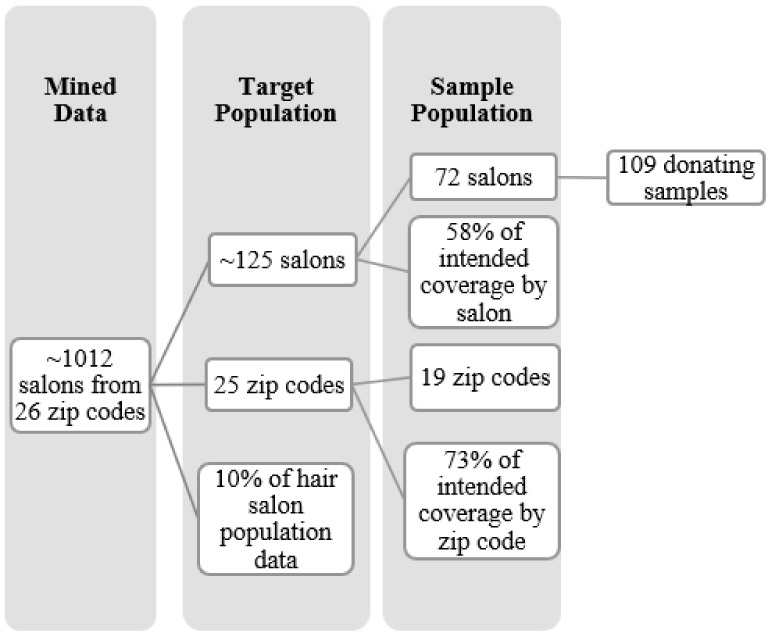
Sampling plan for selecting hair salons by zip code.

**Figure 2 ijerph-19-00086-f002:**
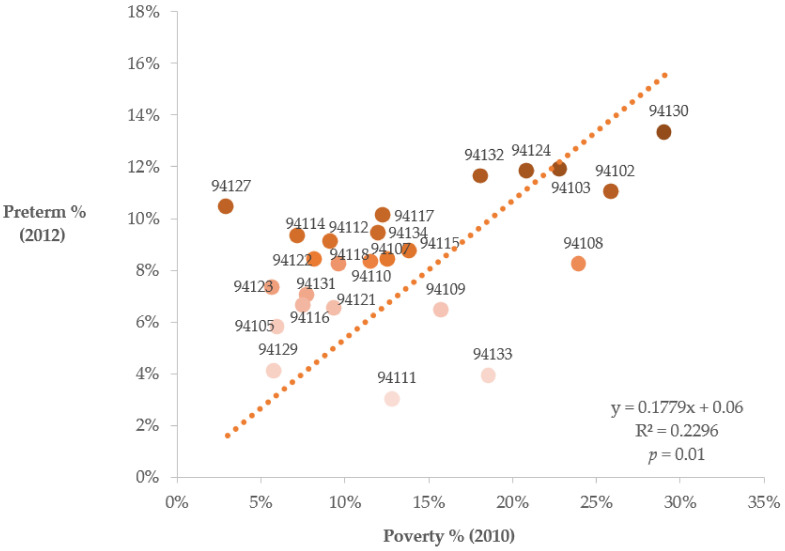
The correlation between percent PTB (*y*-axis) and percent poverty (*x*-axis) by neighborhood (zip code) in SF with a significant *p*-value (0.01) and R^2^ value = 0.23 positive correlation trends. The darker the gradient color, the higher PTB % and Poverty %, *y* = Preterm % (2012), *x* = Poverty % (2010).

**Figure 3 ijerph-19-00086-f003:**
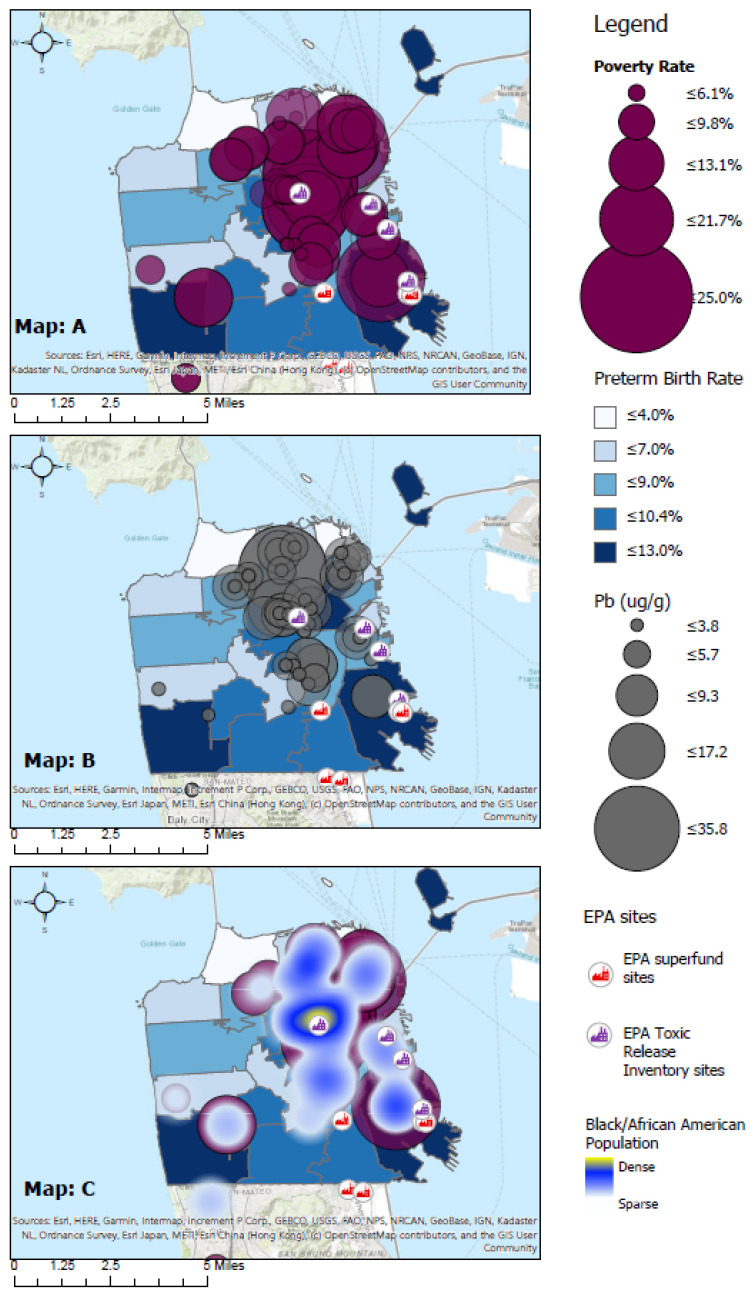
Geographic Distribution of Pb and Social Determinants in San Francisco by zip code. The layers are as follows: a polygon base layer of zip codes with their PTB rate in blue (Maps **A**–**C**), a purple dot distribution of poverty rates (Maps **A**,**C**), a grey dot distribution of Pb concentration Pb (ug/g) by hair salons (Map **B**), and a heat map of African American/Black population density (Map **C**). Black, red, purple icons (Maps **A**–**C**) illustrate EPA sites. Note that zip code 94015 is shared with Daly City and only PTB data exist for Treasure Island), EPA = U.S. Environmental Protection Agency.

**Table 1 ijerph-19-00086-t001:** SF neighborhoods and their zip code referenced.

Zip Code	Neighborhood
94130	Treasure Island
94103	South of Market
94124	Bayview/Hunters Point
94132	Lake Merced/Merced Manor/Lake Shore
94102	North Market/Hayes Valley/Tenderloin
94127	St. Francis Wood/West Portal/Miraloma
94117	Haight-Asbury/Hayes Valley
94134	Visitacion Valley/Portola
94114	Castro/Noe Valley/Corona Heights
94112	Excelsior/Ocean View/Ingleside
94115	Western Addition/Pacific Heights
94107	Potrero Hill
94122	Sunset
94110	Mission/Bernal Heights
94108	Chinatown
94118	Inner Richmond/Presidio
94123	Marina/Cow Hollow
94131	Twin Peaks/Diamond Heights/Glen Park
94158	South of Market/Mission Bay
94116	Parkside/Forest Hill
94121	Richmond/Sea Cliff
94109	Nob Hill/Russian Hill/Tenderloin
94105	Financial District
94129	Presidio
94133	North Beach/Telegraph Hill
94111	Embarcadero

**Table 2 ijerph-19-00086-t002:** Pearson correlations with significant values underneath estimates.

	Pb Concentration	Preterm Birth	Zip Code	African American/Black Population	Foreign-Born	Bachelor’s Degree	Poverty	Gini Index
Pb Concentration	1.0000							
Preterm Birth	0.2749 1.0000	1.0000						
Zip Code	0.0965 0.3204	−0.1168 0.2286	1.0000					
African American/Black Population	0.4023 0.0000	0.4831 0.0000	0.0750 0.4404	1.0000				
Foreign-Born	0.0170 0.8614	0.0649 0.5043	−0.4481 0.0000	0.1428 0.1403	1.0000			
Bachelor’s Degree	−0.2204 0.0219	−0.2775 0.0036	0.2374 0.0134	−0.5340 0.0000	−0.6169 0.0000	1.0000		
Poverty	0.1055 0.2771	0.3511 0.0002	−0.2879 0.0025	0.5216 0.0000	0.5401 0.0000	−0.6169 0.0000	1.0000	
Gini Index	−0.0094 0.9233	−0.0298 0.7592	−0.1786 0.0644	0.1977 0.0402	0.1980 0.0400	−0.2692 0.0048	0.7461 0.0000	1.0000

## Data Availability

More about data availability sources you can find in the Appendix A.

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
