# Peer review of "Geospatial Distributions of Lead Levels Found in Human Hair and Preterm Birth in San Francisco Neighborhoods"

_ijerph, 2021, doi:10.3390/ijerph19010086_

Round 1

Reviewer 1 Report

This Reviewer's concerns have been addressed

Reviewer 2 Report

Thank you for addressing the concerns. All the best.

This manuscript is a resubmission of an earlier submission. The following is a list of the peer review reports and author responses from that submission.

Round 1

Reviewer 1 Report

This study has examined the association between human exposure to lead (Pb) and preterm birth (PTB) in 19 racially diverse SF zip codes. It addresses an important public health issue in this country. Unfortunately, the manuscript is marred by two major concerns: (i) it contains nothing new, and (ii) the study was poorly done.

The lack of science in the paper can be summarized using the primary findings in the Discussion section:

(a). The majority of hair samples obtained from different SF neighborhoods contained Pb. Nothing new scientifically; with a sensitive instrument, one should expect to find lead in any hair sample.

(b). The highest Pb exposure was measured in Eastern SF, predominantly in the Southeast neighborhoods, where a large population of African American resides, and PTB rates soar. This information is not different from the results of a large number of studies under the umbrella of environmental justice.  The only justification for the manuscript therefore is to report the data for the particular study site (SF) – no attempt has been made to address any specific scientific hypothesis.

(c). Additionally, PTB rates were higher in neighborhood pockets encompassing hazardous waste sites, high poverty rates, and racial/ethnic communities. The same has been documented and confirmed in most urban areas of the United States for decades.

Concerns on the study design and protocols include:

(a). The study is grossly underpowered with a sample size of only 109 and so many likely confounding variables

(b). The way the samples were collected leaves a lot to be desired: “Hair samples were collected by research staff filling a 250ml container labeled high-131 density polyethylene with a mixture of trimmed hair from the floor or trash bin”.  What about contamination from the dust and muck on the floor trucked in by customers? The presumption that the data truly represent the hair-Pb for customers from only the ZIP code where the salon is located needs to be validated

(c). “hair was digested with a 1:1 ratio of 100% nitric acid and deionized water to isolate the inorganic metals that accumulates in the hair shaft after environmental exposure”.  Simple acid digestion cannot differentiate the lead from hair shaft from the lead acquired from the shampoo or through the atmosphere  

(d). “Plasma–Atomic Emission Spectrometry (MP-AES) system [was] optimized for measurement of Pb”. This is not a particularly sensitive instrument for measuring the levels of lead in hair (see item below).

(e). “At 405.781 nm, the limit of detection was found to be four parts-per-billion (ppb). Therefore, only measurements of Pb in hair greater than four ppb were deemed reliable and used to determine Pb concentration”. With this criterion, most of the results would be deemed to be unreliable since the reported average concentration is 4.7 µg/g  

This study has examined the association between human exposure to lead (Pb) and preterm birth (PTB) in 19 racially diverse SF zip codes. It addresses an important public health issue in this country. Unfortunately, the manuscript is marred by two major concerns: (i) it contains nothing new, and (ii) the study was poorly done.

The lack of science in the paper can be summarized using the primary findings in the Discussion section:

(a). The majority of hair samples obtained from different SF neighborhoods contained Pb. Nothing new scientifically; with a sensitive instrument, one should expect to find lead in any hair sample.

(b). The highest Pb exposure was measured in Eastern SF, predominantly in the Southeast neighborhoods, where a large population of African American resides, and PTB rates soar. This information is not different from the results of a large number of studies under the umbrella of environmental justice.  The only justification for the manuscript therefore is to report the data for the particular study site (SF) – no attempt has been made to address any specific scientific hypothesis.

(c). Additionally, PTB rates were higher in neighborhood pockets encompassing hazardous waste sites, high poverty rates, and racial/ethnic communities. The same has been documented and confirmed in most urban areas of the United States for decades.

Concerns on the study design and protocols include:

(a). The study is grossly underpowered with a sample size of only 109 and so many likely confounding variables

(b). The way the samples were collected leaves a lot to be desired: “Hair samples were collected by research staff filling a 250ml container labeled high-131 density polyethylene with a mixture of trimmed hair from the floor or trash bin”.  What about contamination from the dust and muck on the floor trucked in by customers? The presumption that the data truly represent the hair-Pb for customers from only the ZIP code where the salon is located needs to be validated

(c). “hair was digested with a 1:1 ratio of 100% nitric acid and deionized water to isolate the inorganic metals that accumulates in the hair shaft after environmental exposure”.  Simple acid digestion cannot differentiate the lead from hair shaft from the lead acquired from the shampoo or through the atmosphere  

(d). “Plasma–Atomic Emission Spectrometry (MP-AES) system [was] optimized for measurement of Pb”. This is not a particularly sensitive instrument for measuring the levels of lead in hair (see item below).

(e). “At 405.781 nm, the limit of detection was found to be four parts-per-billion (ppb). Therefore, only measurements of Pb in hair greater than four ppb were deemed reliable and used to determine Pb concentration”. With this criterion, most of the results would be deemed to be unreliable since the reported average concentration is 4.7 µg/g  

This study has examined the association between human exposure to lead (Pb) and preterm birth (PTB) in 19 racially diverse SF zip codes. It addresses an important public health issue in this country. Unfortunately, the manuscript is marred by two major concerns: (i) it contains nothing new, and (ii) the study was poorly done.

The lack of science in the paper can be summarized using the primary findings in the Discussion section:

(a). The majority of hair samples obtained from different SF neighborhoods contained Pb. Nothing new scientifically; with a sensitive instrument, one should expect to find lead in any hair sample.

(b). The highest Pb exposure was measured in Eastern SF, predominantly in the Southeast neighborhoods, where a large population of African American resides, and PTB rates soar. This information is not different from the results of a large number of studies under the umbrella of environmental justice.  The only justification for the manuscript therefore is to report the data for the particular study site (SF) – no attempt has been made to address any specific scientific hypothesis.

(c). Additionally, PTB rates were higher in neighborhood pockets encompassing hazardous waste sites, high poverty rates, and racial/ethnic communities. The same has been documented and confirmed in most urban areas of the United States for decades.

Concerns on the study design and protocols include:

(a). The study is grossly underpowered with a sample size of only 109 and so many likely confounding variables

(b). The way the samples were collected leaves a lot to be desired: “Hair samples were collected by research staff filling a 250ml container labeled high-131 density polyethylene with a mixture of trimmed hair from the floor or trash bin”.  What about contamination from the dust and muck on the floor trucked in by customers? The presumption that the data truly represent the hair-Pb for customers from only the ZIP code where the salon is located needs to be validated

(c). “hair was digested with a 1:1 ratio of 100% nitric acid and deionized water to isolate the inorganic metals that accumulates in the hair shaft after environmental exposure”.  Simple acid digestion cannot differentiate the lead from hair shaft from the lead acquired from the shampoo or through the atmosphere  

(d). “Plasma–Atomic Emission Spectrometry (MP-AES) system [was] optimized for measurement of Pb”. This is not a particularly sensitive instrument for measuring the levels of lead in hair (see item below).

(e). “At 405.781 nm, the limit of detection was found to be four parts-per-billion (ppb). Therefore, only measurements of Pb in hair greater than four ppb were deemed reliable and used to determine Pb concentration”. With this criterion, most of the results would be deemed to be unreliable since the reported average concentration is 4.7 µg/g  

This study has examined the association between human exposure to lead (Pb) and preterm birth (PTB) in 19 racially diverse SF zip codes. It addresses an important public health issue in this country. Unfortunately, the manuscript is marred by two major concerns: (i) it contains nothing new, and (ii) the study was poorly done.

The lack of science in the paper can be summarized using the primary findings in the Discussion section:

(a). The majority of hair samples obtained from different SF neighborhoods contained Pb. Nothing new scientifically; with a sensitive instrument, one should expect to find lead in any hair sample.

(b). The highest Pb exposure was measured in Eastern SF, predominantly in the Southeast neighborhoods, where a large population of African American resides, and PTB rates soar. This information is not different from the results of a large number of studies under the umbrella of environmental justice.  The only justification for the manuscript therefore is to report the data for the particular study site (SF) – no attempt has been made to address any specific scientific hypothesis.

(c). Additionally, PTB rates were higher in neighborhood pockets encompassing hazardous waste sites, high poverty rates, and racial/ethnic communities. The same has been documented and confirmed in most urban areas of the United States for decades.

Concerns on the study design and protocols include:

(a). The study is grossly underpowered with a sample size of only 109 and so many likely confounding variables

(b). The way the samples were collected leaves a lot to be desired: “Hair samples were collected by research staff filling a 250ml container labeled high-131 density polyethylene with a mixture of trimmed hair from the floor or trash bin”.  What about contamination from the dust and muck on the floor trucked in by customers? The presumption that the data truly represent the hair-Pb for customers from only the ZIP code where the salon is located needs to be validated

(c). “hair was digested with a 1:1 ratio of 100% nitric acid and deionized water to isolate the inorganic metals that accumulates in the hair shaft after environmental exposure”.  Simple acid digestion cannot differentiate the lead from hair shaft from the lead acquired from the shampoo or through the atmosphere  

(d). “Plasma–Atomic Emission Spectrometry (MP-AES) system [was] optimized for measurement of Pb”. This is not a particularly sensitive instrument for measuring the levels of lead in hair (see item below).

(e). “At 405.781 nm, the limit of detection was found to be four parts-per-billion (ppb). Therefore, only measurements of Pb in hair greater than four ppb were deemed reliable and used to determine Pb concentration”. With this criterion, most of the results would be deemed to be unreliable since the reported average concentration is 4.7 µg/g  

Reviewer 2 Report

Review

Geospatial Distributions of Lead Levels Found in Human Hair 2 and Preterm Birth in San Francisco Neighborhoods

Okorie et al.

Overall:

This is a well presented and very relevant manuscript. I applaud the authors for the thoughtful methodology for such a challenging topic.

Some specific concerns I have with regards the methodology:

  1. While the surveys were administered to business owners, did clientele have to consent to their hair being included in samples? Why didn’t clientele have to consent? What is the justification?
  2. Why were surveys administered to only 65 businesses and not 72?
  3. Were the hair samples collected once or several times from each participating business? It seems that collecting the sample over several time points would increase the precision. Especially that this is an ecological study already fraught with some limitations, can you provide the rationale for collecting the hair samples at one time point only (if this was the case).
  4. It is unclear if African American is synonymous with Black. It may be important to qualify especially that some foreign-born individuals may be black. Does Table Supplemental 4 include foreign born black individuals in the Black population?

Results:

  1. I would suggest that Table S3 be included in the main manuscript and not as a supplementary table. If you are restricted by the number of tables and figures you can include in the manuscript, consider moving Table 1 from the main manuscript and place it in the supplementary section and have Table S3 in the main document.
  2. In the correlation assessing the relationship between zip code and any of the variables was zip code considered a continuous variable in that it was plotted in ascending order on the x-axis? How does this affect the interpretation of the results?
  3. Table S5. Model 2 is identical to Model 4. Model 2 should not include the variable % Black Pop.
  4. Consider revising the sentence that appears in lines 227-229. Elaborate on why poverty is not indicative of high PTB rates.

Minor edits:

Line 229: a space is needed between ‘rates’ and ‘than’

In the text, Supplemental Table 3 is referenced before Supplemental Table 2. Consider reordering the tables in the supplementary materials

In table S3, there are two columns labelled ‘Bachelor’s Degree”. I believe the first one of the two should read ‘Zip Code’

Be consistent with Supplement vs. Supplemental
